# MileCut: A Multi-view Truncation Framework for Legal Case Retrieval

Submission Id: 212

## ABSTRACT

In the search process, it is essential to strike a balance between effectiveness and efficiency to improve search experience. Thus, ranking list truncation has become increasingly crucial. Especially in the legal domain, irrelevant cases can severely increase search costs and even compromise the pursuit of legal justice. However, there are truncation challenges that mainly arise from the distinctive structure of legal case documents, where the elements such as fact, reasoning, and judgement in a case serve as different but multi-view texts, which could result in a bad performance if the multi-view texts cannot be well-modeled. Existing approaches are limited due to their inability to handle multi-view elements information and their neglect of semantic interconnections between cases in the ranking list. In this paper, we propose a multi-view truncation framework for legal case retrieval, named MileCut. Mile-Cut employs a case elements extraction module to fully exploit the multi-view information of cases in the ranking list. Then, MileCut applies a multi-view truncation module to select the most informative view and make a more comprehensive cut-off decision, similar to how legal experts look over retrieval results. As a practical evaluation, MileCut is assessed across three datasets, including criminal and civil case retrieval scenarios, and the results show that MileCut outperforms other methods on F1, DCG, and OIE metrics.

## CCS CONCEPTS

• **Information systems** → **Content ranking**; *Learning to rank*; *Top-k retrieval in databases*; **Document filtering**.

## KEYWORDS

Ranking list truncation, Legal case retrieval, Cut-off task, Web Search

**ACM Reference Format:**
Anonymous Author(s). 2024. MileCut: A Multi-view Truncation Framework for Legal Case Retrieval. In *Proceedings of THE WEB CONFERENCE 2024 (WWW'24)*. ACM, New York, NY, USA, 11 pages. https://doi.org/XXXXXXX.XXXXXXX

## 1 INTRODUCTION

Information retrieval, fundamental to web search, is to retrieve documents that are relevant to a query[9, 28]. However, relevant

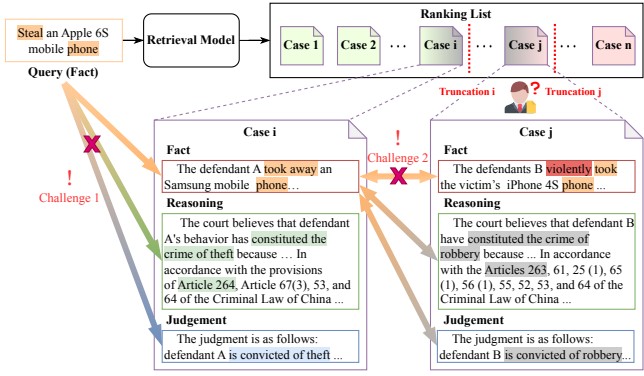

**Figure 1: The challenges in legal cut-off task, where different elements in a case represent isolated views. The darker part represents key content.**

results are inundated with a vast sea of data, hampering user experience in the search process. Thus, it is crucial to implement effective methods to refine the search results by excluding irrelevant or less relevant results from the ranking list. This process is commonly called ranking list truncation or cut-off task[7, 18]. Specifically, ranking list truncation task aims to find an optimal cut-off position in a ranking list that maximizes both efficiency and accuracy.

Different from general web search, the cut-off task is extremely critical in the legal domain, as irrelevant cases in retrieval results can pose significant obstacles. For one thing, irrelevant cases can consume valuable effort. This is because a case document requires careful analysis to determine its applicability and demands more effort compared to a traditional document[22]. For another, irrelevant cases could lead to incorrect conclusions, thereby compromising the pursuit of legal justice since legal analysis requires exacting standards, and any conclusions drawn from inappropriate cases could result in misguided legal strategies. Thus, with the cut-off operation, the quality of retrieval results can be significantly improved. It is beneficial for legal professionals to accomplish their subsequent tasks.

However, several truncation challenges in the legal cut-off task are observed. These challenges stem from the distinctive structure of legal case documents. In detail, a case document includes many elements such as title, procedure, facts, and more. In this work, as illustrated in Figure 1, we consider the three most relevant elements to the retrieval task: fact, reasoning, and judgement. First, these elements provide different yet interconnected descriptions of the case, serving as multi-view text. Particularly in most practical scenarios, the query typically comprises only a colloquial fact description. This will lead to retrieval results primarily based on the fact element, overlooking other crucial elements with significant value for identifying irrelevant results. Secondly, given the complexity and nuance of legal case documents, understanding the relation of

elements between different cases is crucial in efficiently removing irrelevant cases from the search results. For instance, in Figure 1, case i and case j initially seem relevant due to the similarity of their facts to the query. However, they are irrelevant as the reasonings and judgements are vastly different. Thus, it is essential to consider the relation of different views in the ranking list to enhance the precision and efficiency of legal cut-off task.

Thus, it would be intuitive and meaningful to well-design the multi-view texts in a case that can capture the semantics at different granularities. Unfortunately, existing approaches[14, 16, 25, 26] have the following main drawbacks. (1) They are limited to a single view and lack the adaptability to handle multi-view elements of cases in a ranking list. This inherent limitation will hamper the accuracy and effectiveness of predicting the optimal cut-off position. (2) Most truncation methods rely on similarity metrics derived from traditional statistical methods. They ignore capturing the deeper semantic representation of case elements. Especially in a ranking list, it is vital to use the interactive information between cases for identifying relevance. These drawbacks could pose a severe impediment to achieving optimal performance in legal cut-off task.

To address these issues, it is necessary to encode different elements distinctly, capturing comprehensive views from diverse views. Compared to cross encoder, dual encoder is advantageous in accommodating a longer input length, which is vital for extensive legal texts. Additionally, it acquires distinct semantic representations for different elements, accentuating the relationships between various views. Thus, dual encoder excels in differentiating semantic information from multiple views, thereby ensuring a richer and more nuanced representation of the case. Furthermore, the semantic interrelation among different cases within the ranking list is vital. This interconnectedness plays a pivotal role because it enables a more cohesive understanding of the ranking list, promoting a deeper comprehension of the relationship between cases.

In this paper, we propose a novel **M**ulti-v**i**ew truncation framework for **le**gal case retrieval task (named MileCut) by exploiting multi-view information in case document from the ranking list. Specifically, this framework initiates with a query and case documents that pass through a dual encoder, generating embeddings for each element: fact, reasoning, and judgement. MileCut employs a case elements extraction module to capture essential features of different case elements in the ranking list. Then, MileCut uses a multi-view truncation module to identify and incorporate the most informative view into truncation decision-making. Our contributions are summarized as follows:

- We propose a novel multi-view truncation framework for legal case retrieval task that can obtain effective truncation decisions by utilizing multi-view information of cases.
- We first bring the multi-view learning approach into the cut-off task by capturing multi-view features and fusing them into truncation, thereby attaining a comprehensive relation among documents in the ranking list.
- We release a new Chinese Civil Case Retrieval Dataset(C3RD) to the public, facilitating future research on civil case retrieval and cut-off tasks.

Finally, experimental results demonstrate the superior performance of MileCut in the legal cut-off task. The source code is available at https://anonymous.4open.science/r/MileCut-WWW.

## 2 RELATED WORKS

### 2.1 Legal Case Retrieval

Legal case retrieval has been a topic of substantial interest in legal information processing. Especially in recent years, integrating pre-trained language models in the retrieval systems has led to considerable improvements in accuracy and efficiency[23, 29]. For example, Chalkidis et al. [5] explore the application of BERT models to legal tasks and proposed LEGAL-BERT. Xiao et al. [27] develop Lawformer, a Longformer-based model designed explicitly for legal documents, demonstrating the potential of pre-trained models in handling lengthy legal texts. Li et al. [13] propose SAILER, a Structure-Aware pre-traIned language model for LEgal case Retrieval, highlighting the importance of fully utilizing the structural information contained in legal case documents for effective legal case retrieval. Despite the success, these models may inadvertently retrieve irrelevant results, increasing the search effort and even compromising legal justice.

### 2.2 Ranking List Truncation

The ranking list truncation task is an essential part of retrieval systems. Traditional methods[3] for ranking list truncation predominantly relied on strategies based on thresholds or statistics. Recent effective truncation methods are mainly based on deep learning models. For instance, BiCut[14] and Choppy[4] respectively employ Bi-LSTM and Transformer models to capture information in the ranking list and predict optimal cut-off positions. Wu et al. [26] proposed AttnCut, which combines Bi-LSTM and Transformer encoder to capture more comprehensive features. These methods primarily use ranking scores and document statistics as inputs. LeCut[16] takes it a step further by incorporating semantic-level features from the retrieval task, leading to a significant performance improvement. However, the above methods, including LeCut proposed for the legal cut-off task, rely on the general framework and ignore the unique structure of legal case documents. Thus, existing methods fail to account for the multi-view text and would fall short in legal case retrieval scenarios.

## 3 PROBLEM FORMULATION

In this section, we explore the problem definition of the legal cut-off task and clarify preliminary knowledge about legal cases.

The ranking list truncation task aims to find an optimal position for truncating a ranking list from retrieval to balance effectiveness and efficiency. Formally, given a query text $Q$ and a sequence of candidate documents $C=\{c_1, c_2, \cdots, c_n\}(n \in \mathbb{N}^+)$ ranked in decreasing order of relevance, the goal is to predict the optimal truncation position $k \in [1, n]$ that maximizes pre-defined metrics.

To better understand the legal cut-off task, we define various elements in a case document as follows. As illustrated in Figure 1, we focus on the three elements in a case document: fact, reasoning, and judgement. Here, we define the fact element as $F=\{w_i^F\}_{i=1}^{\ell_F}$, the reasoning element as $R=\{w_i^R\}_{i=1}^{\ell_R}$, the judgement element as $J=\{w_i^J\}_{i=1}^{\ell_J}$, where $w$ denotes tokens and $\ell$ denotes the length of the element. Besides, we define the references of law articles as $A=\{a_i\}_{i=1}^{\ell_a}$, where $a$ denotes the article label, and $\ell_a$ is the number of references. In our work, we assume that a query represents the

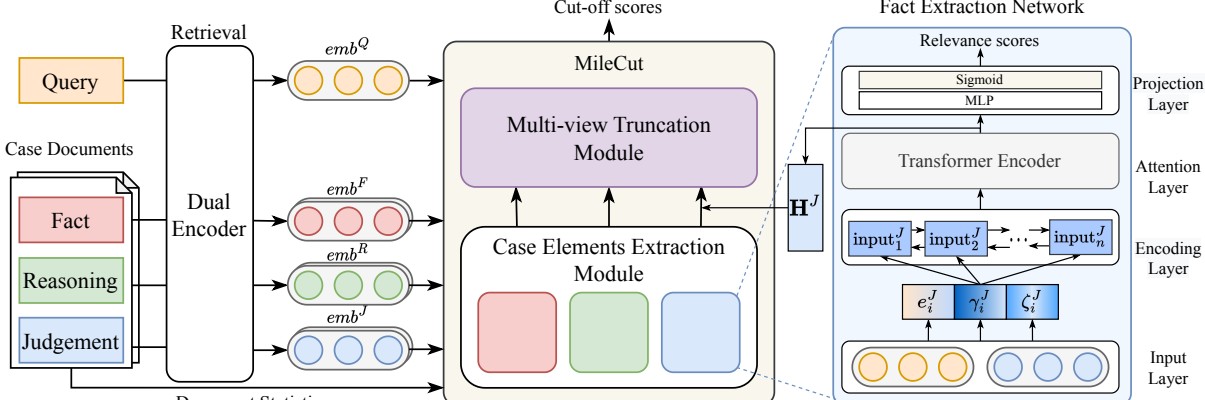

Figure 2: The overall framework of MileCut, which consists of a case elements extraction module and a multi-view truncation module. The dual encoder processes a query and case documents, generating embeddings. These embeddings and document statistics feed into the MileCut to determine cut-off scores. A detailed fact extraction network is illustrated on the right.

basic fact of legal case document. Here, we define the query as $Q=\{w_i^Q\}_{i=1}^{\ell_Q}$ and the case document as $c=\{F, R, J\}$, where $\ell_Q$ is the length of the query.

## 4 MILECUT

### 4.1 Framework Overview

Figure 2 illustrates the overall framework of the MileCut. MileCut takes documents statistics and semantic representations as input and outputs a prediction for the cut-off position. MileCut comprises a case elements extraction module and a multi-view truncation module. The case elements extraction module employs three network structures to extract features of different elements from the ranking list. The multi-view truncation module employs an attention mechanism to select the most informative views from the case elements extraction module and get a more comprehensive representation feature. Finally, MileCut utilizes the fused feature and generates the cut-off scores.

To integrate retrieval information into the cut-off task, both modules employ a similar input layer to process the semantic representation generated by a dual encoder. Besides, these two modules can be trained simultaneously.

### 4.2 Input Preparation and Process

Before truncation, it is necessary to obtain input features from the query and case documents in the ranking list. For the MileCut, these inputs involve document statistics and semantic representations.

Following previous works[14, 16, 26], we adopt the document length, unique number, and statistical similarity as document statistics since they have been proven effective.

For semantic representations, MileCut gets element-level representations which are the outputs of the last hidden layer from a dual encoder. For example, a fact $F$ is transformed into the fact representation as:

$$emb^F=\text{DualEncoder}(F). \quad (1)$$

Similarly, a query, reasoning, and judgement can be given into the corresponding representation $emb^Q$, $emb^R$, and $emb^J$. For the case

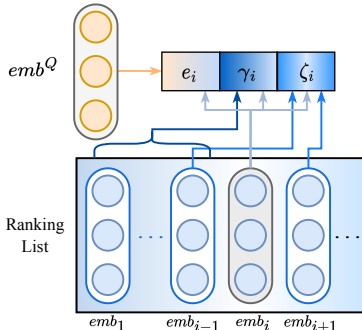

Figure 3: The process of the input layer. Here, $e_i$, $\gamma_i$, $\zeta_i$ respectively denotes the relevance score, precedent similarity score, neighborhood similarity of $i$-th case in ranking list.

document, we use a mean pooling of representations of elements to get the case document-level representation $emb^C$.

To exploit semantic representations, the input layer uses a similar processing method in the case elements extraction module and multi-view truncation module. As shown in Figure 3, the input layer accepts semantic representations and produces semantic listwise features. First, the document-level or element-level similarity score between the query and the $i$-th case is calculated as:

$$e_i = \text{sim}(emb^Q, emb_i), \quad (2)$$

where $emb_i$ can be the document-level or element-level representation of $i$-th case, and sim is a similarity function such as dot product or cosine similarity.

To fuse semantic information into ranking list input, we follow Ma et al. [16] to compute the precedent similarity. Formally, the precedent similarity score computes the similarity score between case representation and its precedents weighted representations in the ranking list as follows:

$$\gamma_i=\text{sim}(emb_i, \sum_{j=1}^{i-1} w_j \cdot emb_j), w_j=\frac{\exp(r_j^C)}{\sum_n^{k=1} \exp(r_k^C)}, \quad (3)$$

where $\gamma_i$ is the precedent similarity score between $i$-th case and its precedents in the ranking list, and $r_i$ is the ranking score of the case.

Moreover, in the legal cut-off task, the interactive information between neighborhood case documents in the ranking list also benefits the truncation position prediction. In contrast to existing methods[16, 26] that compute the cosine similarity of tf-idf or doc2vec between neighborhood documents, we can conveniently compute semantic similarity between representations from the dual encoder. Thus, we further apply document-level and element-level semantic similarity among neighborhood cases as input feature. Specifically, the neighborhood similarity score of a case in the ranking list is given by:

$$
\zeta_i = \begin{cases} \text{sim}(emb_1, emb_2), & i = 0 \\ [\text{sim}(emb_{i-1}, emb_i) + \text{sim}(emb_i, emb_{i+1})]/2, & i \in [2, n) \\ \text{sim}(emb_{n-1}, emb_n), & i = n \end{cases} \tag{4}
$$

where $\zeta_i$ represents the average similarity of $i$-th case to its adjacent cases.

Finally, the semantic features consist of the relevance score, precedent similarity score, and neighborhood similarity score.

## 4.3 Case Elements Extraction Module

The case elements extraction module is designed to extract different semantic features of various elements in the ranking list. There are three extraction networks in the case elements extraction module: fact extraction network, reasoning extraction network, and judgement extraction network. Specifically, the extraction network aims to predict the relevance of case elements according to list-wise relations in the ranking list. For the judgement extraction network, as shown in the right side of Figure 2, there are four layers in the extraction network: 1) Input Layer, 2) Encoding Layer, 3) Attention Layer, and 4) Project Layer.

The input layer computes element-level semantic features according to Equation 2, 3, 4 and then concatenates them as the input: $\text{input}_i^J = \{e_i^J, \gamma_i^J, \zeta_i^J\}$, which of them representing the relevance score, precedent similarity score, neighborhood similarity score.

To better comprehend sequential dependencies and long-range information of each judgement element in the ranking list, we employ a Bi-LSTM as encoder layer and a Transformer[24] encoder as attention layer to encode the input sequences. Formally, given a input sequences $\left\{\text{input}_i^J\right\}_{i=1}^n$, the process is defined by:

$$
\mathbf{M}^J = \text{Bi-LSTM}\left(\left\{\text{input}_i^J\right\}_{i=1}^n\right), \tag{5}
$$

$$
\mathbf{H}^J = \text{Transformer}(\mathbf{M}^J), \tag{6}
$$

where $\mathbf{M}^J \in \mathbb{R}^{N \times n \times m}$ is the output of Bi-LSTM, $\mathbf{H}^J \in \mathbb{R}^{N \times n \times h}$ is the hidden states at the final layer of Transformer Encoder. $N$ indicates the batch size, $m$ and $h$ respectively denote the hidden dimension of the Bi-LSTM and Transformer.

The final hidden state $\mathbf{H}^J$ is used to predict the relevance and deliver the judgement feature to the multi-view truncation module. Specifically, in the projection layer, we apply an MLP layer and a Sigmoid activation layer to obtain the relevance predictions

between the query and each judgement in the ranking list. The formula for projection is as follows:

$$
S^J = \text{Sigmoid}(\text{MLP}(\mathbf{H}^J)) = \{s_i^J\}_{i=1}^n, \tag{7}
$$

where $S^J$ is a list of relevance scores that stands for a relevance probability.

Suppose $Y = \{y_i\}_{i=1}^n$ are the ground-truth labels for candidate cases. Cross-entropy is employed as the loss function of the judgement extraction network, which is defined as follows:

$$
L^J(Y, S^J) = -\sum_{i=1}^n y_i \log(s_i^J). \tag{8}
$$

Similar to judgement extraction network, the reasoning extraction network and judgement extraction network loss functions can be given by:

$$
L^F(Y, S^F) = -\sum_{i=1}^n y_i \log(s_i^F), \tag{9}
$$

$$
L^R(Y, S^R) = -\sum_{i=1}^n y_i \log(s_i^R), \tag{10}
$$

where $S^F$ and $S^R$ are the projection score distribution of the reasoning and judgement extraction networks, respectively. Especially, the input of reasoning extraction network concatenates extra law articles label: $\text{input}_i^R = \{e_i^R, \gamma_i^R, \zeta_i^R, A_i\}$, where $A_i$ is the label of the references law articles.

Finally, we aim to optimize three extraction networks simultaneously, and the overall case elements extraction module loss is the sum of the losses from extraction networks.

## 4.4 Multi-view Truncation Module

The multi-view truncation module is to fuse the features from the case elements extract module and predict the optimal cut-off position. As shown in Figure 4, there are five layers in the truncation network: 1) Input Layer, ) Encoding Layer, 3) Attention Layer, 4) Multi-view Fusion Layer, and 5) Decision Layer.

The input of this module includes ranking scores, document statistics, document-level representations. The input layer computes document-level semantic features and concatenates the document statistics: $\text{input}_i^C = \{r_i, \gamma_i^C, \zeta_i^C, d_i\}$, which of them representing the ranking score, neighborhood similarity score, precedent similarity score, document statistics.

Similar to the case elements extraction module, the encoding layer and attention layer use Bi-LSTM and Transformer encoder to learn the document-level feature. Besides, extra positional embeddings $\mathbf{p}$ are added to the output of Bi-LSTM, serving as the input for the attention layer. The process is calculated as follows:

$$
\mathbf{M}^C = \text{Bi-LSTM}\left(\left\{\text{input}_i^C\right\}_{i=1}^n\right), \tag{11}
$$

$$
\mathbf{H}^C = \text{Transformer}(\mathbf{M}^C + \mathbf{p}), \tag{12}
$$

where $\mathbf{p}$ is a trainable position parameter.

In the multi-view fusion layer, we fuse the hidden states from the case elements extraction module. Since each view feature offers a different informativeness that impacts truncation decisions, we propose a multi-view fusion layer using the attention mechanism to select the most informative feature and get a more comprehensive

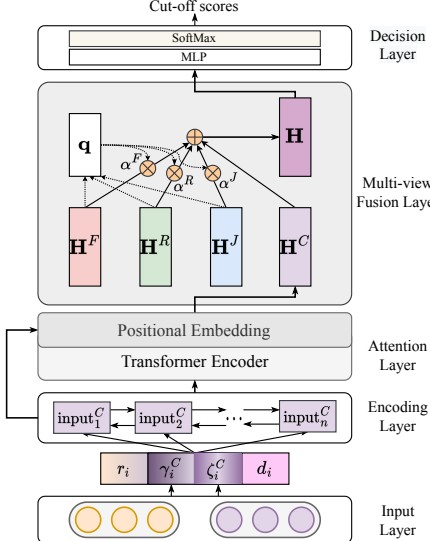

**Figure 4: The structure of the multi-view truncation module. Here, $\mathbf{H}^F$, $\mathbf{H}^R$, and $\mathbf{H}^J$ are external features extracted from the case elements extraction module.**

representation for better truncation. Denote the attention weights of fact, reasoning, judgement as $\alpha^F$, $\alpha^R$, and $\alpha^J$, respectively. The attention weight of the fact view is calculated by:

$$a^F = \mathbf{q}^T \tanh(\mathbf{W} \times \mathbf{H}^F + \mathbf{b}), \tag{13}$$

$$\alpha^F = \frac{\exp(a^F)}{\exp(a^F) + \exp(a^R) + \exp(a^J)}, \tag{14}$$

where $\mathbf{W}$ and $\mathbf{b}$ are learnable attention parameters, $\mathbf{q}$ denotes the attention query vector. The attention weights of reasoning and judgement can be calculated similarly. The final fused multi-view feature is the summation of the truncation transformer layer output and a weighted sum of hidden states from different elements:

$$\mathbf{H} = \mathbf{H}^C + \alpha^F \mathbf{H}^F + \alpha^R \mathbf{H}^R + \alpha^J \mathbf{H}^J, \tag{15}$$

where $\mathbf{H}^F$, $\mathbf{H}^R$, $\mathbf{H}^J$ are the hidden states from the attention layer of the respective element extraction network, and the $\mathbf{H}^C$ is the hidden state from the attention layer of truncation network.

Lastly, an MLP layer and a Softmax activation function are employed to predict the probability distribution of cut-off positions:

$$S^F = \text{Softmax}(\text{MLP}(\mathbf{H})), \tag{16}$$

where $S^T = [s_1^T, s_2^T, \cdots, s_n^T]$ stands for a truncation confidence score of the ranking list.

As denoted in the case elements extraction module, $Y = \{y_i\}_{i=1}^n$ is the ground-truth label for documents, the loss function of multi-view truncation module is defined as:

$$L^T(Y, S^T) = -\sum_{i=1}^n M(Y) \log\left(\frac{\exp(c_i)}{\sum_{j=1}^n \exp(c_j)}\right), \tag{17}$$

where $M$ is the metric score of the truncated ranking list. $M$ could be any truncation metric, such as F1 or DCG.

## 4.5 Training and Inference

The training stage of the case elements extraction module and multi-view truncation module are simultaneous, facilitating the process of joint optimization. We linearly combine the two functions as the overall loss function:

$$\begin{aligned} L &= \lambda L^T + (1-\lambda)L^E \\ &= -\lambda \sum_{i=1}^n M(Y) \log\left[\frac{\exp(c_i)}{\sum_{j=1}^n \exp(c_j)}\right] \\ &\quad + (1-\lambda)\left[-\sum_{i=1}^n y_i \log(s_i^F) - \sum_{i=1}^n y_i \log(s_i^J) - \sum_{i=1}^n y_i \log(s_i^R)\right], \end{aligned} \tag{18}$$

where $L^T$ and $L^E$ are the loss function from the training of case elements extraction module and multi-view truncation module, respectively, and $\lambda \in [0, 1]$ is the coefficient to balance importance between the extraction loss and truncation loss. At the inference stage, for a given ranking list of case documents, the predicted truncation position is determined by the $\text{argmax}(S^T)$.

In summary, the algorithm of MileCut is detailed in Appendix A.

**Table 1: Statistics of datasets. Numbers in brackets denote the size of the training set and test set.**

| Datasets | LeCaRD | C3RD | COLIEE |
|---|---|---|---|
| Language | Chinese | Chinese | English |
| Documents | 43823 | 114600 | 4415 |
| Queries | 107(87/20) | 1146(915/231) | 898(718/180) |
| Candidates/Query | 100 | 100 | 30 |
| Rel Case/Query | 10.33 | 11.43 | 4.67 |

## 5 EXPERIMENTS

### 5.1 Datasets and Metrics

We evaluate MileCut on three legal retrieval datasets: LeCaRD, C3RD, and COLIEE.

- **LeCaRD**[17] is a Chinese legal criminal case retrieval dataset proposed in 2020, and it contains 107 queries and 100 criminal case documents for each query. Following Ma et al. [16], documents labeled 3 or 2 in LeCaRD are regarded as relevant cases.
- **C3RD** is a **C**hinese **C**ivil **C**ase **R**etrieval **D**ataset we build for the case retrieval task. Since the absence of a publicly accessible civil case retrieval dataset, we collect civil case documents from public website and construct C3RD using heuristic rules to fill this void. Further details about the construction of the C3RD dataset can be found in Appendix B. It comprises 1146 queries, each with 100 candidate civil cases documents.
- **COLIEE**[11] is an English legal case retrieval dataset for task 1 in COLIEE 2022. It contains 898 queries and a pool of cases containing 4415 documents, including all the data from previous years. Unlike the other two datasets, each query has to search over 4415 candidate cases for relevant cases. To limit the search space, we randomly select 30 candidates for each query as the dataset.

**Table 2: Comparisons between our method and baselines. The results in bold indicate the best performance on specific datasets.**

| Method | LeCaRD+BERT-Crime | | | C3RD+BERT-Civil | | | COLIEE+RoBERTa | | |
|---|---|---|---|---|---|---|---|---|---|
| | F1 | DCG | OIE | F1 | DCG | OIE | F1 | DCG | OIE |
| Fix@5 | 0.3366 | 2.0629 | 0.1718 | 0.5358 | 1.6950 | 0.1591 | 0.4654 | -0.2266 | 0.1880 |
| Fix@10 | 0.5292 | 3.1158 | 0.2593 | 0.6226 | 1.9226 | 0.2171 | 0.3950 | -1.3055 | 0.1937 |
| Fix@20 | 0.7160 | 4.0664 | 0.3369 | 0.5879 | 1.1082 | 0.2496 | 0.2770 | -3.5041 | 0.1724 |
| Greedy | 0.7438 | 4.2445 | 0.3549 | 0.6226 | 1.9263 | 0.2504 | 0.4678 | 0.0841 | **0.1950** |
| BiCut | 0.7441 | 4.0607 | 0.3531 | 0.6901 | 2.1655 | 0.2533 | 0.4295 | -0.8717 | 0.1933 |
| Choopy | 0.7593 | 4.2214 | 0.3537 | 0.6025 | 1.8963 | 0.1991 | 0.4678 | -0.0546 | 0.1792 |
| AttnCut | 0.7273 | 4.0790 | 0.3414 | 0.7621 | 2.9370 | 0.2503 | 0.4656 | 0.1253 | 0.1727 |
| MtCut | 0.7109 | 4.0502 | 0.3352 | 0.7635 | 2.9592 | 0.2525 | 0.4733 | 0.1253 | 0.1776 |
| LeCut | 0.7594 | 4.3469 | 0.3538 | 0.7532 | 2.8522 | 0.2504 | 0.4689 | 0.1165 | 0.1765 |
| MileCut | **0.7835** | **4.6335** | **0.3562** | **0.7767** | **3.0623** | **0.2533** | **0.5044** | **0.2154** | 0.1877 |
| Oracle | 0.8530 | 5.3858 | 0.3601 | 0.8813 | 3.6169 | 0.2665 | 0.7023 | 1.1526 | 0.2121 |

| Method | LeCaRD+BERT-Chinese | | | C3RD+BERT-Chinese | | | COLIEE+BERT-base | | |
|---|---|---|---|---|---|---|---|---|---|
| | F1 | DCG | OIE | F1 | DCG | OIE | F1 | DCG | OIE |
| Fix@5 | 0.3405 | 2.0427 | 0.1716 | 0.5255 | 1.6429 | 0.1573 | 0.4373 | -0.3314 | 0.1790 |
| Fix@10 | 0.5087 | 2.8777 | 0.2513 | 0.6124 | 1.8415 | 0.2143 | 0.3914 | -1.3441 | 0.1876 |
| Fix@20 | 0.6983 | 3.8288 | 0.3291 | 0.5847 | 1.0496 | 0.2476 | 0.2764 | -3.5427 | 0.1674 |
| Greedy | 0.7459 | 3.8737 | 0.3493 | 0.6124 | 1.8652 | 0.2491 | 0.4373 | 0.0181 | **0.1881** |
| BiCut | 0.7483 | 4.0465 | 0.3474 | 0.6635 | 1.9487 | 0.2502 | 0.4153 | -0.9462 | 0.1874 |
| Choopy | 0.7582 | 4.1344 | 0.3480 | 0.5982 | 1.8639 | 0.1980 | 0.4320 | -0.1637 | 0.1693 |
| AttnCut | 0.7185 | 4.0987 | 0.3341 | 0.7477 | 2.8378 | 0.2477 | 0.4383 | 0.0576 | 0.1681 |
| MtCut | 0.7184 | 4.1988 | 0.3351 | 0.7336 | 2.7375 | 0.2524 | 0.4319 | -0.0445 | 0.1661 |
| LeCut | 0.7588 | 4.1522 | 0.3480 | 0.7154 | 2.6263 | 0.2450 | 0.4416 | 0.0223 | 0.1732 |
| MileCut | **0.7684** | **4.3529** | **0.3494** | **0.7508** | **2.8871** | **0.2490** | **0.4764** | **0.1336** | 0.1802 |
| Oracle | 0.8447 | 5.2355 | 0.3536 | 0.8623 | 3.5373 | 0.2643 | 0.6830 | 1.1416 | 0.2054 |

The detailed statistics of three datasets are shown in Tabel 1.

For each retrieval dataset, we adopt the pre-trained Transformer model as the dual encoder. In detail, BERT[8] and RoBERTa[15] are adopted to COLIEE as they are pre-trained on English corpus. Then BERT-Crime and BERT-Civil[30] are adopted to LeCaRD and C3RD, respectively, as they are pre-trained on Chinese criminal and civil cases, respectively. In addition, BERT-Chinese is applied in LeCaRD and C3RD. Finally, the cut-off datasets are named according to the combination of retrieval dataset and dual encoder.

As for evaluation measures, F1 at rank k (F1@k) and Discounted Cumulative Gain at rank k (DCG@k)[10] are employed to evaluate the performance of all methods following previous works[16, 25, 26]. Additionally, Observational Information Effectiveness(OIE)[1] – a metric based on Shannon's information theory - is adopted as OIE is one of the best candidates for truncated ranking evaluation[2].

## 5.2 Baselines and Experimental Setup

Following truncation methods are employed as baselines:

- **Fix@k** set a fixed cut-off position k in a ranking list, after which all other results are therefore truncated.
- **Greedy** finds a fixed cut-off position k based on the training set and uses it for the test set.
- **BiCut**[14] is an RNN-based model along with a flexible cost function to determine the optimal cut-off point.
- **Choopy**[4] adopts a Transformer encoder to find the optimal cut-off position, requiring only the relevance scores.

- **AttnCut**[26] employs both Bi-LSTM and Transformer architecture to solve the cut-off task effectively.
- **MtCut**[25] adopts the multi-gate mechanism and mixture-of-experts approach to incorporate two auxiliary tasks, ensuring a fair and accurate cut-off process.
- **LeCut**[16] captures semantic features from the retrieval task to make better cut-off decisions specifically designed for the legal cut-off task.
- **Oracle** always selects the optimal cut-off position, providing the upper limit performance in terms of the chosen evaluation metrics.

We implement our proposed method in PyTorch[19]. The details about the experimental setup can be found in Appendix C.

## 5.3 Experimental Results

The main results can be found in Table 2. We can obtain the following observations: First, it is evident that most neural methods overall outperform traditional methods. The inherent flexibility of neural models allows them to dynamically truncate a ranking list, rather than determining a fixed cut-off position. Notably, while Greedy appears to have a good result, particularly on the OIE metric, its efficacy is largely contingent upon the alignment of training and test set distributions. If there's a discrepancy in the distribution of oracle cut-off positions between the two sets, the performance of Greedy is likely to deteriorate. Secondly, we can find that LeCut and MileCut surpass other methods in most results.

Table 3: Ablation study on the LeCaRD and C3RD datasets. "w/o F/R/J" represents removing fact/reasoning/judgement extraction network. "w/o attn" represents removing attention mechanism from the multi-view fusion layer. "w/o $\zeta$" represents removing the semantic neighborhood similarity from the input layer.

| Method | LeCaRD+BERT-Crime | | | C3RD+BERT-Civil | | |
|---|---|---|---|---|---|---|
| | F1 | DCG | OIE | F1 | DCG | OIE |
| MileCut | **0.7835** | **4.6335** | **0.3562** | **0.7767** | **3.0623** | **0.2533** |
| w/o F | 0.7735 (↓ 1.3%) | 4.5251 (↓ 2.3%) | 0.3551(↓ 0.3%) | 0.7354 (↓ 5.3%) | 2.8498 (↓ 6.9%) | 0.2401 (↓ 5.2%) |
| w/o R | 0.7829 (↓ 0.1%) | 4.6123 (↓ 0.5%) | 0.3561(↓ 0.0%) | 0.7785 (↑ 0.2%) | 3.0607 (↓ 0.1%) | 0.2530 (↓ 0.1%) |
| w/o J | 0.7814 (↓ 0.3%) | 4.6145 (↓ 0.4%) | 0.3562(− 0.0%) | 0.7778 (↑ 0.1%) | 3.0465 (↓ 0.5%) | 0.2524 (↓ 0.3%) |
| w/o attn | 0.7829 (↓ 0.1%) | 4.6901 (↑ 1.2%) | 0.3563(↑ 0.0%) | 0.7746 (↓ 0.3%) | 3.0111 (↓ 1.7%) | 0.2538 (↑ 0.2%) |
| w/o $\zeta$ | 0.7755 (↓ 1.0%) | 4.6185 (↓ 0.3%) | 0.3547(↓ 0.4%) | 0.7672 (↓ 1.2%) | 3.0008 (↓ 2.0%) | 0.2526 (↓ 0.3%) |

It shows that semantic information from the retrieval task is vital for better truncation. Although Choppy surpasses LeCut in certain results, Choppy exhibits instability due to its sole reliance on ranking scores, disregarding the uncertainty of the retrieval model[6]. Thirdly, compared with other competitors, MileCut achieves the best performance in LeCaRD and C3RD datasets, both F1, DCG, and OIE metrics, indicating the effectiveness of our method. Especially in the C3RD dataset, MileCut outperforms other methods significantly. This mainly benefits from its novel utilization of multi-view information extracted from case documents, considerably enhancing the result of legal cut-off task. In addition, even though all neural methods are trained to maximize the F1 score, most of them still yield competitive results across other metrics. A possible reason is that F1 could effectively satisfy properties for cut-off task, including confidence, recall, and redundancy. These properties are directly related to the truncation position[2]. Furthermore, even in the COLIEE dataset, MileCut generally outperforms other neural methods, despite making truncation decisions solely on document-level information. This suggests that the improvement achieved by semantic neighborhood similarity is significant.

## 5.4 Ablation Studies

To demonstrate the effectiveness of each module in MileCut, we create five variants by selectively removing each component of our framework. Specifically, for the case elements extraction module, we remove three different extraction networks individually. For the multi-view truncation module, we remove the attention mechanism in the multi-view fusion layer and directly add hidden states to obtain the fused feature. Furthermore, we remove the semantic neighborhood similarity from the input layer of both modules. We conduct the ablation study on the LeCaRD and C3RD datasets, using BERT-Chinese as the dual encoder. The results can be found in Table 3.

We can obtain the following conclusions: First, the truncation performance decreases without any of the three extraction networks, proving the effectiveness of the case elements extraction module. This result reveals that utilizing additional features of different elements such as reasoning can help the disentanglement of irrelevant cases in ranking list. Notably, the absence of the fact extraction network has the most significant impact, indicating that the fact information is particularly crucial in the truncation process. Secondly, the performance generally decreases without the attention mechanism, showing that the multi-view fusion layer indeed

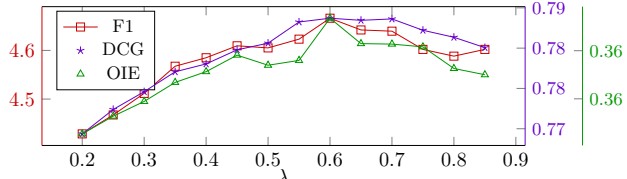

Figure 5: The impact of the scale parameter $\lambda$ selection on the "LeCaRD+BERT-Crime" dataset.

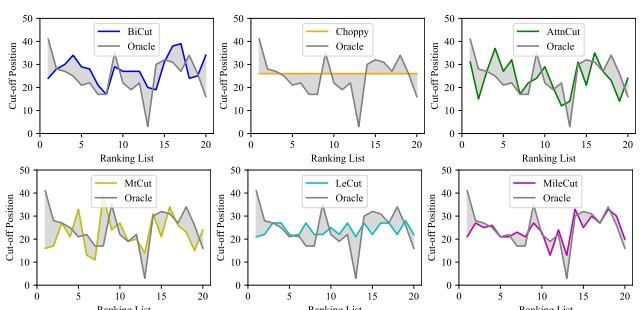

Figure 6: The cut-off positions on the "LeCaRD +BERT-Chinese" test set. The horizontal axis represents different ranking lists and the vertical axis represents cut-off positions. The smaller the shadow area means the better performance.

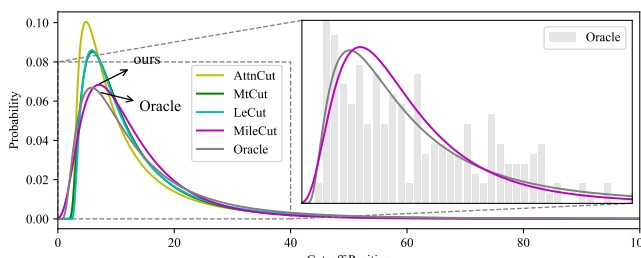

Figure 7: The distribution of cut-off positions on the "C3RD+BERT-Chinese" test set. All lines are log-normal distributions fitted to the positions. The closer to Oracle means the better performance.

works. Lastly, we can observe that the truncation performance decreases without $\zeta$, which means semantic neighborhood similarity is vital to filter out the irrelevant cases in a ranking list.

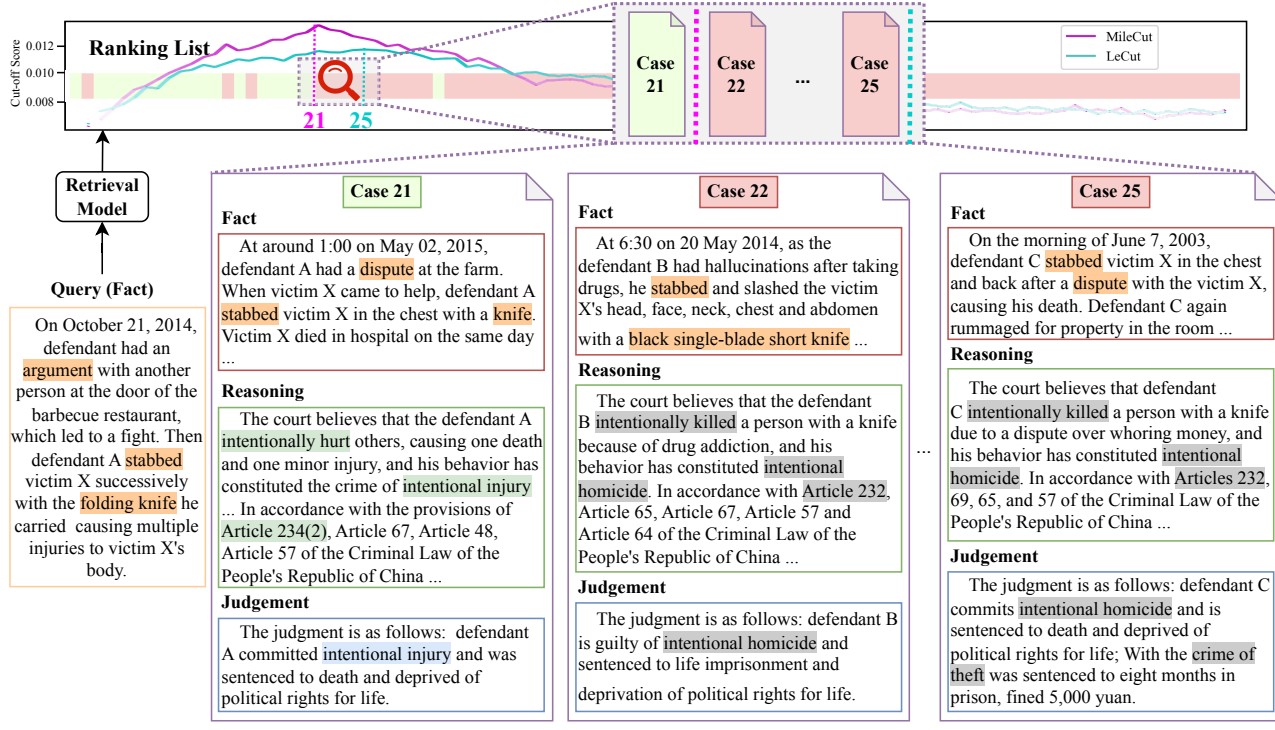

**Figure 8: An example of MileCut. The ranking list belongs to query 2132 in the "LeCaRD+BERT-Crime" dataset. In the upper section, the horizontal axis represents cut-off positions, and the vertical axis represents the cut-off scores. Cases marked in green signify relevant cases. The darker part means key content.**

In the training stage, the impact of the coefficient $\lambda$ selection in the loss function is shown in Figure 5. We can find that a value between 0.6 and 0.7 generally achieves overall better performance. This suggests that both two modules are crucial, and the performance is more effective when these two modules are appropriately balanced. In practice, we adopt $\lambda$ as 0.6 to ensure a good balance between the contributions of both modules.

## 5.5 Case Study

Figure 6 shows the differences of cut-off positions between MileCut and other competitive baselines on the "LeCaRD+BERT-Chinese" test set. MileCut best approximates the Oracle cut-off strategy, indicating that MileCut can capture multi-view features in case documents from the ranking list, thus bringing better cut-off positions. Besides, although LeCut has comparable results, it lacks the capability to predict more accurate cut-off positions. Figure 7 provides a more intuitive depiction of the effectiveness of MileCut. Here, the log-normal distribution is employed to model the distribution of cut-off positions. It is evident that MileCut aligns more closely with the distribution of Oracle.

Figure 8 displays a cut-off example wherein various cases are ranked according to their relevance to a query. The top of the figure provides a visual comparison of the cut-off scores between MileCut and LeCut. In this example, MileCut determines a cut-off position at case 21, and LeCut extends the cut-off position to case 25. Specifically, case 21 is identified as a relevant case. While irrelevant cases 22 and 25 attain a higher rank predominantly as the similarity of their facts to the query, the reasonings and judgements diverge markedly. The crux of this distinction lies in the intentions of defendants. The defendants in case 22 and 25 possessed a deliberate intent to deprive another individual of life, instead of injurement.

From the curve of cut-off scores, LeCut seems to lean heavily on factual similarities and does not capture this nuanced difference behind facts. It gives similar cut-off scores to cases from 22 to 25 based primarily on their factual similarities to the query. This lack of discernment in capturing the intricate features leads to an inaccurate cut-off at position 25. Conversely, MileCut catches the subtle yet critical difference in case 22 from multi-view information, leading to a drastic drop in the cut-off score for position 22 and achieving the optimal cut-off at position 21. This result underscores MileCut effectively captures the multi-view features in the ranking list, leading to more precise truncation results.

## 6 CONCLUSION AND FUTURE WORK

This paper proposed a novel multi-view truncation framework for legal retrieval task named MileCut. By considering the multi-view information and fusing them into cut-off decision-making, MileCut captures a comprehensive semantic feature of case elements in the ranking list. Experimental results demonstrate that MileCut outperforms other competitors in the legal cut-off task. In addition, the effectiveness of MileCut is further confirmed by an ablation study. In future work, the multi-view consideration could be expanded to more elements. Besides, this strategy could be applied to other domains involving multi-view text data.

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

## A MILECUT ALGORITHM

Algorithm 1 describes the full implementation of the proposed MileCut.

## B CHINESE CIVIL CASE RETRIEVAL DATASET

Civil case retrieval presents unique challenges and opportunities. First, civil cases outnumber criminal cases, indicating a significant demand for their retrieval in real-world applications. Secondly, civil cases are inherently more complex to retrieve civil cases. Unlike the clearer facts in criminal cases, civil cases often have muddled facts, as both parties emphasize favorable aspects. Last but not least, there are no publicly available datasets for civil case retrieval. Hence, our proposed **C**hinese **C**ivil **C**ase **R**etrieval **D**ataset(C3RD) aims to fill this void.

To construct the C3RD dataset, we collect over 23 million civil case documents from China Judgements Online website[1], a resource published by the Supreme People's Court of China. Next, we proceeded to refine the corpus by applying a filtering process. The intent was to exclude cases that might be considered too brief or excessively lengthy. In addition, we discarded cases that had been withdrawn in order to focus on cases that had proceeded to full legal resolution. After pre-processing, 8 million civil case documents are left. For retrieval purposes, we developed a criterion to identify relevant cases. According to a guidance document about relevant case retrieval published by the Supreme People's Court of China[2], a relevant case is defined as a case that shares similarities with a query case in aspects such as facts, cause reason and application of law articles. Based on this guidance, we design heuristic rules to filter cases related to the query. Specifically, we deem cases with the same legal cause reason and references to specific law articles as relevant cases. We then randomly select the fact section of a case to serve as a query and remove that case from the pool of relevant candidates. Lastly, we adopted BM25 to search for negative candidates to complete the candidate pools. In this process, we

---

[1] https://wenshu.court.gov.cn
[2] Guidance Opinions on Unifying the Application of Law and Strengthening Similar Case Retrieval by the Supreme People's Court (Trial Implementation)

---

**Algorithm 1:** MileCut

---

**Input** : Case document ranking list $C = \{(c_i)\}_{i=1}^{n}$, document statistics $D = \{(d_i)\}_{i=1}^{n}$, Semantic representations $emb^Q$, $\{(emb_i^F)\}_{i=1}^{n}$, $\{(emb_i^R)\}_{i=1}^{n}$, $\{(emb_i^J)\}_{i=1}^{n}$, Law articles labels $\{A_i\}_{i=1}^{n}$.

**Output** : The cut-off position $p \in [1, n]$.

1 Initialize MileCut parameters, including case elements extraction module $\Theta_E$ and multi-view truncation module $\Theta_T$.

  **for** $c_i$ in $C$ **do**

    // Input process

2     Calculate $\text{input}_i^F$, $\text{input}_i^R$, $\text{input}_i^J$; // {Eq.2, 3, 4}

3     Calculate $\text{input}_i^C$. // {Eq.2, 3, 4}

  **end**

  **while** *MileCut has not converged* **do**

    // Case elements extraction module

4     Calculate hidden states $\mathbf{H}^F$, $\mathbf{H}^R$, $\mathbf{H}^J$; // {Eq. 5}

5     Project the relevance scores $S^F$, $S^R$, $S^J$; // {Eq. 16}

6     Calculate $L^F$, $L^R$, $L^J$; // {Eq. 9, 10, 8}

7     $L^E = L^F + L^R + L^J$;

    // Multi-view truncation module

8     Calculate hidden state $\mathbf{H}^T$; // {Eq. 5}

9     Calculate attention weights $\alpha^F$, $\alpha^R$, $\alpha^J$; // {Eq. 13, 14}

10     Fuse multi-view hidden states $\mathbf{H}$; // {Eq. 15}

11     Predict the cut-off score $S^T$; // {Eq. 16}

12     $p = \text{argmax}(S^T)$;

13     Calculate $L^T$; // {Eq. 17}

14     Calculate final loss $L$; // {Eq. 18}

15     Update parameters of two modules $\Theta_E$, $\Theta_T$.

  **end**

  **return** $p$

---

apply several filtering methods to ensure the identified cases aren't related to the query. These measures are put in place to maximize the likelihood that the selected cases differ considerably from the query. Finally, C3RD comprises 1146 queries, and each query has 100 candidate civil case documents.

For evaluation, we implement several existing retrieval models on C3RD as baselines. This will provide a comprehensive view of the characteristics and its applicability for different retrieval methods. The results are shown in Table 4

## C EXPERIMENTAL SETUPS

For the retrieval task, we utilize the Sentence-Transformer[20] to train a dual encoder, which is specifically a siamese dual encoder with shared parameters. All of the retrieval models adopt the default hyper-parameter settings, and their dimensions of embeddings are 768. Moreover, the document-level similarity $e_i^C$ is adopted as the ranking score $r_i$. Then, all documents are then ranked in descending order of ranking score.

For the cut-off task, we apply BM25[21] similarity as statistical similarity in both MileCut and other baselines. Both BM25[3] and baselines[4] are derived from open-source implementations, with

---

[3]https://github.com/dorianbrown/rank_bm25

[4]https://github.com/Woody5962/Ranked-List-Truncation

---

their hyper-parameters set to default values. For OIE metric, we set $\beta$ to 1.05 and $\mathfrak{D}$ to the length of ranking list.

For LeCaRD dataset processing, we extract the references of law articles. For each case, we select the first referred article in chapter two as the article label because chapter two of Criminal Law details crimes. The label is formatted as {chapter number, section number, article number}. For C3RD dataset processing, we select the top-5 referred articles as article labels because a civil case typically includes references to multiple laws. The label is formatted as {law name, article number}. Specifically, for the COLIEE dataset, MileCut employs only the truncation module due to the case documents in the COLIEE dataset are difficult to segment into element-level texts.

For MileCut, we specially set the dimension of the Transformer last hidden layer in two modules to be the same. We determine the learning rate as 3e-5, dropout as 0.2, and the hidden size of Bi-LSTM and Transformer encoder as 128 and 256, respectively.

For training, $\lambda$ is set to 0.6, and the batch size is set as 8, 20, and 20 for LeCaRD, C3RD, and COLIEE, respectively. Adam[12] is adopted as the optimization algorithm.

## D EXTEND METRIC FOR EVALUATION

DCG is an appropriate measure for comparing the results of the same query. However, the maximum achievable DCG value can vary across queries. Therefore, the performance of a query with

**Table 4: Evaluation of baseline models on C3RD. "dual" and "cross" denote dual encoder and cross encoder, respectively.**

| Method | P@5 | P@10 | MAP@10 | NDCG@10 | NDCG@20 | NDCG@30 | MRR |
|---|---|---|---|---|---|---|---|
| BM25 | 0.5079 | 0.4146 | 0.4835 | 0.5642 | 0.5810 | 0.5993 | 0.6929 |
| BERT-Civil(dual, w/o training) | 0.5670 | 0.4773 | 0.7419 | 0.6235 | 0.6528 | 0.6841 | 0.7455 |
| BERT-Chinese(dual) | 0.7108 | 0.6231 | 0.6999 | 0.7786 | 0.8208 | 0.8451 | 0.8149 |
| BERT-Civil(dual) | 0.7732 | 0.6736 | 0.7905 | 0.8546 | 0.8813 | 0.8978 | 0.8788 |
| BERT-Civil(cross) | 0.7609 | 0.6355 | 0.7682 | 0.8406 | 0.8669 | 0.8863 | 0.9137 |

**Table 5: The nDCG@Oracle results.**

| Method | LeCaRD+BERT-Crime | C3RD+BERT-Civil | COLIEE+BERT-RoBERTa |
|---|---|---|---|
| Fix@5 | 0.3619 | 0.5086 | 0.2806 |
| Fix@20 | 0.5227 | 0.3831 | 0.0406 |
| Greedy | 0.6824 | 0.5086 | 0.3378 |
| BiCut | 0.6788 | 0.5681 | 0.2116 |
| Choopy | 0.7115 | 0.5119 | 0.3140 |
| AttnCut | 0.6783 | 0.7161 | 0.3418 |
| MtCut | 0.6870 | 0.7320 | 0.3312 |
| LeCut | 0.7184 | 0.6957 | 0.3289 |
| MileCut | **0.7560** | **0.7511** | **0.3685** |
| Oracle | 1.0000 | 1.0000 | 1.0000 |

**Table 6: Extend analysis on the LeCaRD datasets.**

| Method | LeCaRD+BERT-Crime | | | C3RD+BERT-Civil | | |
|---|---|---|---|---|---|---|
| | F1 | DCG | OIE | F1 | DCG | OIE |
| MileCut | **0.7835** | **4.6335** | **0.3562** | **0.7767** | **3.0623** | **0.2533** |
| w/o $A$ | 0.7817 ($\downarrow$ 0.2%) | 4.6636 ($\uparrow$ 0.6%) | 0.3549 ($\downarrow$ 0.4%) | 0.7784 ($\uparrow$ 0.2%) | 3.0613 ($\downarrow$ 0.0%) | 0.2529 ($\downarrow$ 0.1%) |

a larger number of relevant articles affects a lot compared to the one with a smaller number of relevant articles. [10] propose to use n(normalized)DCG to remove such effects. In typical retrieval tasks, nDCG is indeed useful as the ideal DCG is derived from the optimal possible ranking list. However, in the cut-off task, where the ranking list is fixed, nDCG doesn't fit. To address such effects, we conduct additional experiments. We use the DCG of Oracle as the 'ideal' DCG (instead of DCG value from the best ranking list) to normalize the DCG values at different truncation positions for each query. This isn't the ordinal nDCG, and we tentatively name it nDCG@Oracle.

## E  EXTEND ANALYSIS FOR MILECUT

We further examine the influence of including article labels as a part of the input to the reasoning extraction network. Thus, we remove the label $A$ of the "LeCaRD" dataset, and the results are shown in Table 6. The results indicated that the label does not significantly enhance the results. That may be because differences in versions of the Criminal Law of China could potentially lead to inconsistencies when labeling. Furthermore, a legal case may reference multiple law articles, making it challenging to assign a unique, impactful label. Future studies could look into incorporating techniques for handling such issues.

