# OpenReview forum: "MileCut: A Multi-view Truncation Framework for Legal Case Retrieval"
_ACM.org/TheWebConf/2024/Conference — TheWebConf24 Oral_

### Official Review · Reviewer_4beh · 2023-10-24

**Novelty:** 5
**Technical Quality:** 5

**Review:**

In this paper, the authors address the challenge of optimizing the search process, particularly in the legal domain, where irrelevant cases can significantly impact search costs and the pursuit of legal justice. The primary aim of the paper is to improve ranking list truncation, a critical task that seeks to strike a balance between search efficiency and accuracy. This is of the utmost importance in the legal domain, where the distinctive structure of legal case documents, which contain multiple elements such as fact, reasoning, and judgment, is a significant challenge.

To address these challenges, the paper proposes a multi-view truncation framework named "MileCut." MileCut utilizes a dual encoder to capture comprehensive representations of different case elements (fact, reasoning, and judgment) in the ranking list. It then employs a multi-view truncation module to make more informed cut-off decisions by identifying and incorporating the most informative view. This approach outperforms other methods in common metrics, as demonstrated through practical evaluations on three datasets related to criminal and civil case retrieval scenarios.

PROS
(+) Existing approaches are limited in their ability to handle multi-view elements and neglect the semantic interconnections between cases in the ranking list. In this paper, the authors try to address that limitation;
(+) The paper confirms the effectiveness of MileCut in the legal cut-off task across three datasets, including criminal and civil case retrieval scenarios. It reports improvements in F1, DCG, and OIE metrics, demonstrating that MileCut outperforms other methods;
(+) It introduces MileCut, a novel framework for improving ranking list truncation in the legal domain by leveraging multi-view information which outperforms existing methods in legal case retrieval tasks. This innovative approach has the potential to significantly improve the efficiency and effectiveness of legal case retrieval;
(+) a new dataset related to China civil cases is released encompassing 8 million civil case documents posing itself as an important resource for further research.

CONS
(-) Lack of Detailed Experimental Statistical Analysis: The paper lacks a comprehensive analysis of the experimental results. While it claims that MileCut outperforms other methods on F1, DCG, and OIE metrics, it does not provide detailed statistics when comparing it with baselines. This lack of concrete evidence makes it challenging to assess the significance of the proposed system's improvements.
(-) Unclear Model Complexity: The paper does not thoroughly discuss the model's complexity and resource requirements. It would be beneficial to understand how computationally intensive the dual encoder and other components are, especially when applied to large legal databases. This information is crucial for practical implementation.
(-) Inadequate Discussion of Limitations: The paper mentions the challenges of handling multi-view elements and the legal cut-off task's unique structure. However, it does not delve into the limitations and potential failure scenarios of MileCut. A more comprehensive discussion of when and why the system may not perform well is needed.
(-) Figures Description: The figures provided in the paper are essential for understanding the architecture, but not always understandable or properly explained in the text. I find it challenging to grasp the system's components and their interactions due to this.
(-) Ethical issues: a new dataset was proposed. However, ethical issues were not discussed.

All in all the paper has the potential to make a significant contribution to the field of legal case retrieval. The innovative approach and detailed framework make it a promising candidate for acceptance. However, there are areas that require revision and improvement. For instance, the paper lacks a comprehensive analysis of experimental results with insufficient concrete statistical evidence to support the claim of MileCut's superiority over other methods. Furthermore, the complexity and resource requirements of the system need more thorough discussion, particularly when applied (in the future) to large legal databases. The paper should also include a more comprehensive discussion of its limitations and potential failure scenarios.

**Questions:**

In Section 4.2 the authors mention using document statistics, including "unique number." What is the definition of "unique number" in this context? Also, how is "statistical similarity" defined and computed for the documents? Are there variations in these features that the authors experimented with, and what impact did they have on the model's performance?

Dual encoders are a flexible and effective approach for a wide range of NLP tasks where understanding relevance is crucial. In Information Retrieval, they are often used to improve the relevance of search results (by encoding both the user query and the documents) to quickly identify relevant documents. In this paper, a dual encoder process is applied by the authors in their MILECUT architecture. Yet, the paper lacks details on the architecture, such as the dimensionality of the embeddings, and why a dual encoder was chosen over other models.

The authors also refer to a document-level representation emb^C, however, such representation is not illustrated in Fig 2

The source and nature of the ranking list used for the cut-off task are also not adequately explained. It would be helpful to understand how this list is generated.

It was also not clear to me how the authors preprocess the legal documents, including the three key elements: fact, reasoning, and judgment.

In section 4.4 the authors refer to positional embeddings being added to the output of Bi-LSTM. However, I have not seen a study of their significance in the context of the model has not been presented.

In the Multi-View Truncation Module, the paper mentions fusing the multi-view feature using an attention mechanism. Could the authors provide more details on how this fusion works and how it benefits the truncation decision-making?

For the inference stage, how does the model predict the truncation position based on the output scores? Is it a simple argmax operation, or is there a more complex decision-making process involved?

Could the authors discuss the trade-off between model complexity and performance? Was there a consideration of using simpler models or architectures to achieve similar results, and if so, what were the outcomes?

How does the proposed MileCut system scale with larger datasets (with millions of cases), and to what extent is it generalizable beyond the specific legal domain? Are there any computational or resource limitations that need to be considered?

What ethical considerations and permissions were taken into account when extracting data from "https://wenshu.court.gov.cn"? Specifically, how did the authors address concerns related to data privacy, copyright, and the terms of use for the website when creating this dataset for research purposes?

**Reviewer Confidence:**

2: The reviewer is willing to defend the evaluation, but it is likely that the reviewer did not understand parts of the paper

**Scope:**

4: The work is relevant to the Web and to the track, and is of broad interest to the community

---

### Official Review · Reviewer_mooH · 2023-11-12

**Novelty:** 5
**Technical Quality:** 4

**Review:**

This paper proposes an innovative approach to address the challenges of ranking list truncation in legal case retrieval. It introduces MileCut, a framework that utilizes a multi-view truncation method, focusing on the elements of fact, reasoning, and judgment in legal documents. This approach is designed to enhance the precision and efficiency of legal case retrieval by effectively managing the multi-view nature of legal texts and their interconnections.

**Strengths**
1. The approach of multi-view truncation is a significant advancement in the domain of legal case retrieval, addressing the unique challenges posed by the multi-faceted nature of legal documents.
2. The inclusion of a case elements extraction module and a multi-view truncation module demonstrates a thorough understanding of the intricacies involved in legal case retrieval.
3. The release of a new Chinese Civil Case Retrieval Dataset (C3RD) is commendable, as it facilitates future research in this area.

**Weaknesses**
1. The complexity of the model might limit its practical application, especially in environments where resources are constrained.
2. In the Case Elements Extraction Module, the three extraction networks appear to lack differentiation and have not been specifically designed for targeted purposes.
3. Some experimental studies on different model architectures (other than Bi-LSTM+Transformer) are lacking.

**Questions:**

1. Could the authors elaborate on the potential computational costs and efficiency of implementing MileCut in a real-world scenario?
2. How interpretable is the MileCut model in terms of explaining its truncation decisions in a legal context? Given the critical nature of legal case retrieval, can practitioners easily understand why certain cases are included or excluded in the results?

**Reviewer Confidence:**

2: The reviewer is willing to defend the evaluation, but it is likely that the reviewer did not understand parts of the paper

**Scope:**

3: The work is somewhat relevant to the Web and to the track, and is of narrow interest to a sub-community

---

### Official Review · Reviewer_xzuz · 2023-11-20

**Novelty:** 4
**Technical Quality:** 5

**Review:**

This paper focuses on the task of ranking list truncation for legal case
retrieval, and proposes a multi-view truncation framework to tackle this problem.

Pros
* This paper is well-written and easy to follow.

* This paper is evaluated on three datasets, covering both Chinese and English.

* The proposed method is compared with many baselines, and some improvement can be observed over the baselines.

* The code for this paper is publicly available, which facilitates reproducibility.

Cons
* There is one concern for the setup of the ranking list truncation task, which aims to find an optimal position to truncate the ranking list. This task assumes that the ranking list already gives the optimal ranking and only a cutoff is needed. A better setup might be first reranking the results, followed by truncating the reranked list. In this way, we can make sure that lower ranked documents from retrieval can still be included.

* Some baselines perform very closely to the proposed method in Table 2. Please add statistical significance test.

**Questions:**

Will the C3RD dataset be made public?

**Reviewer Confidence:**

3: The reviewer is confident but not certain that the evaluation is correct

**Scope:**

4: The work is relevant to the Web and to the track, and is of broad interest to the community

---

### Official Review · Reviewer_oWWm · 2023-11-22

**Novelty:** 4
**Technical Quality:** 5

**Review:**

The paper focuses on the legal domain, particularly on legal case retrieval, where the authors propose an
approach to address the task of result list truncation. The approach leverages a multi-view
representation of legal documents, focusing on three document legal elements (fact, reasoning, and
judgment). The paper introduces a Chinese civil case retrieval dataset as a second contribution.
Specific Comments
- The introduction effectively outlines the task, its motivation, and the limitations of existing
methods.
- The authors compare their approach with several approaches in the literature.
- Line 110-111: The authors claim, "Practically in most practical scenarios, the query typically
comprises only a colloquial fact description"; however, this claim is not justified by a related
study or evidence presented in this work.
- Moreover, in lines 231 and 252, the authors state: "In our work, we assume that a query
represents the basic fact of legal case document"; However, in the COLIEE task 1 dataset, a
given query is a whole legal case. Therefore, it is expected to contain fact, reasoning, and
judgment elements, contradicting the paper's assumption.
- Reproducing the proposed approach is difficult based solely on its description in Section 4.
However, the availability of the source code somehow neglects this issue.
- The authors have not performed statistical significance testing in the presented results. In
addition, in Table 2, there are some errors in boldface, e.g. C3RD+BERT-Chinese, Greedy vs.
MileCut.
- Based on the ablation study, the contribution of the several components of the framework is
inconsistent. Nonetheless, statistical testing can help clarify whether these observations are
significant.
- The author&#39;s decision to leverage 30 randomly selected cases for each query of the COLIEE
dataset should be further elaborated. It would be essential that the authors justify why it is
needed to limit the search space.
- As the approach needs a query, i.e., it is performed online, it would be valuable that the authors
comment on the method&#39;s efficacy. If the approach is too slow, a greedy approach with the same
OIE measure may be more appropriate for real-world applications.
Overall, the problem addressed in the study is intriguing, and the proposed approach appears
promising. Nonetheless, additional details on its evaluation would have been beneficial to assess the
potential impact and utility of the framework.

COMMENT BASED ON THE AUTHORS' FEEDBACK
Thank you for the provided clarifications. Your work addressing the previously mentioned limitations by revising and reproducing parts of the work has significantly enhanced the paper's value. In light of these improvements, I will update my scores to reflect the enhanced significance of the work.

**Questions:**

Could the authors provide more insight into your decision to use only 30 randomly selected cases for
each query in the COLIEE dataset? It would be helpful to understand the rationale behind limiting the
search space in this manner, and how this choice impacts the study&#39;s findings and their
generalizability.

In lines 231 and 252, the authors mention that a query represents the basic fact of a legal case
document. However, the COLIEE task 1 dataset typically includes queries that encompass the entire
legal case, including facts, reasoning, and judgment. Could the authors clarify how this aligns with
your paper&#39;s assumption, and discuss any implications this might have on your approach and
results?

**Ethics Review Description:**

-

**Reviewer Confidence:**

3: The reviewer is confident but not certain that the evaluation is correct

**Scope:**

4: The work is relevant to the Web and to the track, and is of broad interest to the community

---

### Official Review · Reviewer_ncf4 · 2023-11-29

**Novelty:** 3
**Technical Quality:** 5

**Review:**

This paper argues that it is important to consider multiple views of legal cases in truncating the ranking list in legal case retrieval. The proposed MileCut model first utilizes a dual encoder to obtain the representation for each view and then combines the multi-view information to find an optimal cut-off position in the ranking list.

Strengths:
+ A new dataset will be released.
+ Extensive ablation study.

Weaknesses:
- The novelty is somewhat limited as the list truncation problem and the multi-vector representation of long legal documents have been extensively investigated in the existing literature.
- The introduction of the problem and the description of the proposed model can be further improved. I'd recommend adding a brief introduction of the evaluation metrics in Section 3, which would help readers fully understand the actions we need to choose and the trade-offs we need to consider in the list truncation problem. Some parts of section 4 are also hard to follow. For example, why the module introduced in 4.3 is a "case elements extraction module" as it just a module that tries to fit Y with the information in each view?
- No significance test for the comparisons in Table 2.

**Questions:**

1. In Eqn. 5 and 6, you stack a transformer layer upon a Bi-LSTM. Why not just use a two-layer transformer or two-layer Bi-LSTM here?

2. How did you extract the facts, reasoning, and judgment from case documents? Did you use different encoders for these three views? If you use a shared encoder here, is it possible to have a more fine-grained representation of the case documents by using each paragraph as a view?

3. Did you also consider different views in the query case? From Figure 2, it seems that the query is represented as a single embedding. Why not also use a multi-view representation for the query case?

**Reviewer Confidence:**

3: The reviewer is confident but not certain that the evaluation is correct

**Scope:**

3: The work is somewhat relevant to the Web and to the track, and is of narrow interest to a sub-community

---

### Decision · Program_Chairs · 2024-01-22

**Decision:**

Accept (Oral)

**Comment:**

This is the meta-review. The paper proposes MileCut, a multi-view truncation framework for legal case retrieval that addresses the challenges of ranking list truncation. A dual encoder is utilized to capture comprehensive representations of different case elements. Evaluations on three datasets have been conducted to verify the effectiveness of the proposed framework. Comparative studies on multiple baselines have been performed.

 During the discussion phase, the authors provided thorough information. Additional experimental results have been added. Some major and common concerns of the original reviews have been clarified clearly, such as the statistical significance test, the model complexity, and efficiency, the ethics issue, etc. More details on experimental settings have been given.
 All reviewers feel that most of the questions have been answered and are satisfied with the discussions.

 Therefore, generally speaking, after discussion and the improvement of the submission, the paper is well-written; the framework has some novel modules; and the experimental results are reliable; the code is online available, which will help make the work reproducible. Furthermore, the authors promised to release the new dataset, which will contribute to the community.

 Although there is still some work and questions left for future work, given no paper is perfect, I think now the paper is ready for publication, if the authors will add these improvements during the discussions into the final version.